# HYPOTHESIS-TEST DRIVEN COORDINATE ASCENT FOR REINFORCEMENT LEARNING

## ABSTRACT

This work develops a novel black box optimization technique for learning policies for stochastic environments. Through combining coordinate ascent with hypothesis testing, Hypothesis-test Driven Coordinate Ascent (HDCA) optimizes without computing or estimating gradients. The simplicity of this approach allows it to excel in a distributed setting; its implementation provides an interesting alternative to many state-of-the-art methods for common reinforcement learning environments. HDCA was evaluated on various problems from the MuJoCo physics simulator and OpenAI Gym framework, achieving equivalent or superior results to standard RL benchmarks.

## 1 INTRODUCTION

In recent years, Reinforcement Learning (RL) techniques have contributed to striking advancements in artificial intelligence. From mastering complicated games like Go (Silver et al., 2017) to autonomous helicopter flight (Ng et al., 2006), RL agents continue to make impressive strides. RL is primarily concerned with discovering a policy that encodes a mapping from states to actions that optimizes rewards within a given environment. This work proposes a method for learning policies in stochastic environments where agent performance is evaluated via a black box function.

Much of RL focuses on Markov Decision Processes (MDPs), where an agent's actions are chosen to maximize the expected value of a reward function over time (Russell & Norvig, 2019). Due to the inherent variability within real-world problem domains, the outcome of these actions depends on the action taken and noise within the environment. MDPs treat time as a discrete quantity, where actions are chosen at each time step based on a learned policy that leads the agent to a new state. Different states correspond to specific rewards, which are used to refine the policy parameters. Typically, parameter updates are taken in the direction that maximizes the expected reward and often involve calculating (or approximating) gradients.

The function of interest is opaque for black box optimization (BBO); only point-wise evaluations are possible and the true gradient is inaccessible. As a result, gradient-based techniques must approximate the true gradient through function sampling and calculating mean and variance estimates. Calculating reliable estimates can be difficult in practice and is made especially challenging by the stochastic nature of MDPs. While various gradient-based BBO methods have found success (Mania et al., 2018; Salimans et al., 2017; Bibi et al., 2020), they may converge to local optima and can be slow to converge for complex domains. As an alternative to gradient approximations, we propose Hypothesis-test Driven Coordinate Ascent (HDCA), which leverages the use of derivative-free optimization (DFO) techniques operating on a black box function.

*Our first major insight* is to leverage coordinate ascent to decompose the large search space of the original policy optimization problem into a sequence of sub-problems with smaller search spaces (Wright, 2015). By considering simple enough sub-problems, we are able to optimize each iterate using a basic random search with uniform sampling from a defined search range. This process involves generating batches of candidate solutions at each iteration and updating the policy to the best performing candidate. While there are numerous methods to explore the high-dimensional policy space, other more informed search policies, such as Importance Sampling (Bibi et al., 2020), involve significant computational overhead as well as additional reliance on approximated gradients. Uniform sampling avoids these complications and allows us to take full advantage of modern advances in parallel and distributed computing. While sufficiently exploring the perturbation space

may require generating several candidate solutions, each model evaluation may be done entirely in parallel of the other candidates. Therefore, with appropriate computational resources, our method can converge in fewer iterations than other current methods.

*Our second major insight* is to account for the stochastic nature of our optimization problem. That is, the cumulative reward is a stochastic quantity whose distribution depends on the stochastic policy and environment. Thus, one must estimate the expected cumulative reward with finite samples of rollouts. This leads to variance and uncertainty about true expectations of cumulative rewards for candidate solutions. The stochastic nature (and finite sample estimation) of the cumulative reward criterion *has been largely ignored by previous BBO approaches*, where the mean over rollouts is (incorrectly) taken as the ground truth criterion. In practice this may result in updates that happen to perform better by chance, rather than due to policy improvement. Here, we develop a principled approach that accounts for stochasticity with one-sided, two-sample hypothesis tests to update policy parameters based on statistical significance.

Lastly, we perform several experiments and ablations to empirically study HDCA. When evaluated on a sample of RL control problems, we found HDCA to consistently converge to a solution in fewer iterations than other comparable methods. Given HDCA's highly scalable nature, the reduced number of iterations required to find a solution translates to quicker training time given adequate computational resources.

## 2 METHODS

### 2.1 MARKOV DECISION PROCESS

An MDP describes an environment where an agent acts within a fully observable, stochastic domain to maximize its expected cumulative discounted reward (Russell & Norvig, 2019). The environment can be described as a set of states, $S$, along with a set of actions, $A$, that can be taken to transition between states. The transition function $p(s'|s, a)$ describes the probability that taking action $a$ in state $s$ leads to state $s'$, and the reward function $R(s, a)$ describes the reward received for taking action $a$ in state $s$.

A solution to an MDP is represented as a policy $\pi_\theta$, encoded by parameters $\theta$, that maps states to a distribution over actions in a way that maximizes accumulated rewards. This paper focuses on environments where the reward function acts as a black box.

### 2.2 COORDINATE ASCENT

Our first insight is to use coordinate ascent to decompose the problem of optimizing the large policy parameter space into a sequence of smaller, more feasible, sub-problems. Coordinate ascent maximizes a function $f(\theta)$ by successively maximizing sub-problems with respect to varying a single coordinate at a time. Similarly, block coordinate ascent iterates through blocks of coordinates and updates a subset of multiple coordinates sequentially (Tseng, 2001) (see Algorithm 1).

Choose $\theta^0 \in \mathbb{R}^d$;
Set $i \leftarrow 0$;
**repeat**
 Given $\theta^i$, choose a block of coordinates $b$ in the set $\{1, ..., d\}$ and compute new iterate $\theta^{i+1}$
  that satisfies;

$$\theta_b^{i+1} \in \arg\max_{\theta_b} f(\theta_p, \theta_b) \tag{1}$$

$$\theta_p^{i+1} = \theta_p^i, \forall p \notin b \tag{2}$$

 $i \leftarrow i + 1$;
**until** *termination test satisfied*;

**Algorithm 1:** Block Coordinate Ascent

We approximate coordinate ascent optimization to maximize policy parameters $\theta$ without gradients. To do so, we numerically optimize the sub-problem for a block of coordinates, rather than analyt-

ically solving it. If the dimension of the block of coordinates being updated is small, even simple numerical optimization techniques yield good estimates of optimal updates. We propose approximate block coordinate ascent (ABCA), which simply performs a random search over the coordinates of the block that is being maximized (see Algorithm 2).

The primary benefit of utilizing ABCA is that optimization occurs without the use of gradient approximation, which can be challenging even for deterministic functions. When applied to reinforcement learning, the difficulties are exacerbated by the stochastic dynamics of many real-world RL problems, which lead to noisy function evaluations and rough approximations of the gradient. When combined with a potentially non-convex reward function, following an inaccurate gradient can lead to sub-optimal solutions and slow convergence. ABCA permits us to avoid gradient estimation altogether and instead make updates using simple global numerical techniques that are feasible for the smaller sub-problems.

We can see from Algorithm 2 that the update rule simply takes the policy corresponding to the highest empirical average cumulative reward within each batch of perturbed $\theta$ values for a block of coordinates. This differs from typical gradient estimation, where the update rule is implemented as a weighted sum across the batch of perturbed coordinates. For example, in the popular Evolutionary Strategies approach discussed in Sec. 3, updates are computed via an *average* (Eq. 12) over perturbations. Due to the concentration of empirical means, this results in diminishing returns as one increases the number of perturbations per update (iteration). In contrast, ABCA considers a *maximum* over perturbations, and thus is able to see greater returns as we increase the number of perturbations considered per update. As one may compute trajectories, $t_m^{(j)}$, and cumulative rewards, $\mathbf{R}(t_m^{(j)})$, in parallel, the better utility of perturbations implies that ABCA is able to take better advantage of the benefits of distributed computing compared to gradient estimation techniques.

**Input:** Initial policy parameters $\theta^0 \in \mathbb{R}^d$;
**Hyperparameters**: Search range $s$, number of noise samples $n$, Number of coordinates per block $c$, number of rollouts in mean calculation $M$;
Set $i \leftarrow 0$;
**repeat**

Let coordinate block $b$ consist of $c$ randomly selected coordinates from the set $\{1, ..., d\}$;
Sample $\epsilon_{j,k} \sim \mathrm{U}(s)$ **for** $j = 1, ..., n$, and $k \in b$;
Let $\hat{\theta}^{(j)}$ be the $j^{\mathrm{th}}$ candidate optimizer (perturbation) over block $b$ with values:

$$\forall q \in \{1, \ldots, d\} \setminus b, \ \hat{\theta}_q^{(j)} \leftarrow \theta_q^i, \quad \forall k \in b, \ \hat{\theta}_k^{(j)} \leftarrow \theta_k^i + \epsilon_{j,k} \tag{3}$$

Using the policy samples $a_i \sim \pi_{\hat{\theta}^{(j)}}(s_i)$ and environment samples $s_{i+1} \sim p(s_{i+1} \mid s_i, a_i)$, generate $M$ rollouts $t_m^{(j)}$, and compute respective cumulative rewards $\mathbf{R}(t_m^{(j)})$, $\forall m \in \{1, \ldots, M\}$:

$$t_m^{(j)} \leftarrow \left\{ (s_{m,0}^{(j)}, a_{m,0}^{(j)}), (s_{m,1}^{(j)}, a_{m,1}^{(j)}), \ldots \right\} \quad \mathbf{R}(t_m^{(j)}) = \sum_{(s,a) \in t_m^{(j)}} R(s,a) \tag{4}$$

Update $\theta$ based on the empirical average cumulative rewards $\bar{\mathbf{R}}(\hat{\theta}^{(j)}) \leftarrow \frac{1}{M} \sum_{m=1}^{M} \mathbf{R}(t_m^{(j)})$:

$$\mathbf{j} \leftarrow \arg\max_j \bar{\mathbf{R}}(\hat{\theta}^{(j)}), \quad \theta^{i+1} \leftarrow \hat{\theta}^{(\mathbf{j})} \tag{5}$$

$i \leftarrow i + 1$;

**until** *termination test satisfied*;

**Algorithm 2:** Approximate Block Coordinate Ascent (ABCA)

## 2.3 Hypothesis testing

Our second insight is to note that, unlike deterministic optimization, the stochastic nature of our optimization problem (through the environment dynamics and stochastic policy) requires a careful consideration of variance over expected rewards when comparing different perturbations in ABCA

iterates. That is, as we are enumerating random policies, it may be due to chance, rather than an improvement in parameters, that the average reward for a candidate optimizer perturbation is greater. Below we propose to account for the stochasticity of expected reward approximations using hypothesis testing.

Given the variance in the sequence of states encountered during policy rollouts, the cumulative reward can be highly variable, and thus selecting the policy corresponding to the empirical maximum reward may not lead to improved overall performance. To generate a more accurate assessment of a given model's expected reward, we make multiple function evaluations for each perturbed $\hat{\theta}^{(j)}$ (see Eq. 4) and compute corresponding mean and variance estimates from the set of policy rollouts:

$$\bar{\mathbf{R}}(\hat{\theta}^{(j)}) \leftarrow \frac{1}{M} \sum_{m=1}^{M} \mathbf{R}(t_m^{(j)}), \quad \sigma^2_{\mathbf{R}(\hat{\theta}^{(j)})} \leftarrow \frac{1}{M} \sum_{m=1}^{M} (\mathbf{R}(t_m^{(j)}) - \bar{\mathbf{R}}(\hat{\theta}^{(j)}))^2. \tag{6}$$

We are ultimately interested in determining if there is a perturbed model that is statistically demonstrated to yield a higher expected reward than our current model, thus a one-sided, two-sample t-test is best suited for this task (NIST/SEMATECH, 2003). Since our policy perturbations at any given iteration are selected from a uniform distribution, it is likely that some of the perturbed models will lead to worse overall performance. Additionally, since function evaluations represent independent environment rollouts, the data in our two-sample t-test is unpaired. Operating under the likelihood that the variances of the two samples are not equivalent, the test statistic, $T$, for the two-sample $t$-test is calculated as: $T = (\mu_1 - \mu_2)/\sqrt{\sigma_1^2/N_1 + \sigma_2^2/N_2}$ where $\mu_i$, $\sigma_i$, and $N_i$ correspond to the $i^{th}$ sample's mean, standard deviation, and sample size, respectively (Cressie & Whitford, 1986).

In our method, we seek to compare the rewards of candidate policy parameters $\hat{\theta}^{(j)}$ to the rewards of the current iterate $\theta^i$; thus, test statistics are derived from the respective means and variances for reward:

$$T_j = \frac{\bar{\mathbf{R}}(\hat{\theta}^{(j)}) - \bar{\mathbf{R}}(\theta^i)}{\sqrt{\sigma^2_{\mathbf{R}(\hat{\theta}^{(j)})}/M + \sigma^2_{\mathbf{R}(\theta^i)}/M}} \tag{7}$$

The null hypothesis $H_0$ for this test is that the expected performance of the current policy is greater than or equal to the perturbed policy that is being compared. Since we are only interested in models with higher expected rewards, we reject the null hypothesis if

$$T_j > t_{1-\alpha,\nu_j}, \tag{8}$$

where $t$ corresponds to the critical value of the $t$ distribution with significance threshold $\alpha$ and $\nu_j$ degrees of freedom, calculated as

$$\nu_j = \frac{\left(\sigma^2_{\mathbf{R}(\hat{\theta}^{(j)})}/M + \sigma^2_{\mathbf{R}(\theta^i)}/M\right)^2}{(\sigma^2_{\mathbf{R}(\hat{\theta}^{(j)})}/M)^2/(M-1) + (\sigma^2_{\mathbf{R}(\theta^i)}/M)^2/(M-1)}. \tag{9}$$

Finally, the update step concludes by updating the model to the perturbed $\hat{\theta}$ with the highest degree of statistical evidence to outperform the previous iteration's model, if such a model is found (or passes through the previous iterate, otherwise). Thus, HDCA considers a test-based update, which replaces the update in ABCA (Eq. 5) with:

$$\mathbf{j} \leftarrow \arg\max_j T_j - t_{1-\alpha,\nu_j}, \quad \theta^{i+1} \leftarrow \hat{\theta}^{(\mathbf{j})} \text{ if } T_{\mathbf{j}} - t_{1-\alpha,\nu_{\mathbf{j}}} > 0, \text{ else } \theta^i. \tag{10}$$

That is, HDCA implements ABCA with an update that is based on statistical significance for better policy updates (see appendix for full statement of HDCA algorithm). It is important to note that the two-sample t-test has a parameter for the significance level $\alpha$ that is likely to be problem specific. In the realm of hypothesis testing, this parameter controls whether or not the null hypothesis gets rejected. When applied to our optimization scheme, this parameter determines if a policy demonstrates enough statistical evidence to be considered a more optimal policy than the previous iteration. Therefore, this parameter can be used to control the degree of policy exploration, with a larger $\alpha$ indicating a higher affinity to update the policy to a perturbed value.

## 3 RELATED WORK

### BLACK BOX OPTIMIZATION

This work uses the term "black box" as a qualifier for objective functions in which only point-wise evaluations are possible, producing a single scalar output for each input. To maximize this objective function, the problem can be written as $\max_{\theta \in R^d} = F(\theta)$, where $\theta$ represents the d-dimensional input parameters (Choromanski et al., 2020). Since the internal mechanisms governing the objective are hidden, this form of optimization is derivative-free. Therefore, typical solutions are iterative in nature, where perturbations to $\theta$ occur at the beginning of each round of training. Such solutions are known as parameter-perturbation methods (Stulp & Sigaud, 2012). Within RL, these parameters encode a policy that maps states to actions. The function evaluation corresponds to the total reward for a policy rollout, or trajectory, so information about the specific states encountered during the rollout is not available to the agent.

BBO algorithms have demonstrated success in common RL benchmarks, such as the gradient-estimation techniques ES (Salimans et al., 2017) and Augmented Random Search (ARS) (Mania et al., 2018) as well as gradient-free techniques like GradientLess Descent (GLD) (Golovin et al., 2019). HDCA separates itself from these comparable techniques by sampling multiple trajectories per perturbed policy, and thus is able to integrate the variance in the estimated expected cumulative reward when making policy updates to optimize in a gradient-free manner.

**Evolutionary Strategies.** In the ES approach proposed by Salimans et al. (2017), an objective function $F$ acts on candidate solutions defined by parameters $\theta$, where $\theta$ is distributed according to some distribution $p_\psi$. ES, then, is concerned with maximizing $\mathbb{E}_{\theta \sim p_\psi} F(\theta)$. Letting $p_\psi$ be a multivariate Gaussian distribution with mean $\psi$ and variance $\sigma^2 I$, this optimization problem can be written as $E_{\theta \sim p_\psi} F(\theta) = E_{\epsilon \sim N(0,I)} F(\theta + \sigma\epsilon)$. In order to optimize this expected value, ES implements stochastic gradient ascent, with gradient steps taken based on the estimation:

$$\nabla \mathbb{E}_{\epsilon \sim N(0,I)} F(\theta + \sigma\epsilon) = \frac{1}{\sigma} \mathbb{E}_{\epsilon \sim N(0,I)} F(\theta + \sigma\epsilon)\epsilon. \tag{11}$$

For each candidate solution within a given iteration $t$, $F_n = F(\theta^t + \sigma\epsilon_n)$ is evaluated and the parameter update step is defined as:

$$\theta^{t+1} = \theta^t + \alpha \frac{1}{N\sigma} \sum_{n=1}^{N} F_n \epsilon_n, \tag{12}$$

where $\alpha$ is the learning rate and $N$ is the population size.

### MEAN-VARIANCE OPTIMIZATION

Several proposed RL methods incorporate mean-variance estimates to train risk-sensitive agents. Xie et al. (2018) explore this concept in their Mean-Variance Policy Gradient (MVP) implementation. Rather than simply maximizing the expected cumulative reward, MVP aims to learn the best performing policy that keeps the variance in the estimated cumulative reward below some threshold value. MVP reformulates the mean-variance trade-off function as its Legendre-Fenchel dual, and subsequently uses cyclic block coordinate ascent to maximize this objective. While MVP shares similar inspiration with HDCA, MVP is not a black-box optimization technique, nor is it gradient-free. Further, despite both methods incorporating the variance in the expected return to guide policy updates, MVP uses this variance as an upper-bound, while HDCA uses the variance in its hypothesis testing framework for performance comparison across policy perturbations.

Alternatively, Thomas et al. (2015) propose High Confidence Policy Improvement as a framework for generating new policies with probabilistic guarantees. This approach asks the user to supply a performance floor $\rho_-$ and confidence level $\delta$, and the algorithm will return a policy that has a lower expected cumulative return than $\rho_-$ with probability at most $\delta$. The interpretation of $\delta$ is closely related to $\alpha$ used in HDCA's hypothesis tests, with both being dependent on how much risk is reasonable for the given task. High Confidence Policy Improvement, however, is designed as a framework for ensuring policy safety and is not concerned with how proposed policies are trained and generated. It is therefore not a black-box optimization technique, unlike HDCA.

## 4 EVALUATION

Experiments were conducted on multiple RL environments to analyze the training performance of HDCA as compared to other blackbox optimization methods: ARS (Mania et al., 2018), ES (Salimans et al., 2017), and GLD (Golovin et al., 2019). We compare performance based on the number of parameter updates (iterations), which are the bottleneck when training with simulated environments and parallel computing. Sec. 4.1 analyzes performance on the LunarLanderContinuous-v2 environment, where HDCA and ES train a neural network and ARS trains a linear model. Sec. 4.2 explores the performance of HDCA, ARS, and GLD on various MuJoCo locomotion tasks, where every approach trains identically structured linear policies.

**Implementation details.** Our implementation of HDCA is provided in the supplemental material. We use the Pathos multiprocessing Python framework (McKerns et al., 2012) to implement a parallel version of Algorithm 3. Since the majority of HDCA's computational overhead derives from evaluating the reward function, we created a pool of workers to compute episode rollouts in parallel. We evaluated HDCA on various MuJoCo locomotion tasks (Todorov et al., 2012) and the LunarLanderContinuous-v2 environment simulated through the OpenAI Gym (Brockman et al., 2016). The OpenAI Gym provides benchmark reward functions for all simulated environments. For each environment, experiments were run until a threshold performance level was reached averaged over 100 trials.

### 4.1 LUNARLANDERCONTINUOUS-V2

In this section, we compare the training performance of HDCA, ES, and ARS for the LunarLanderContinous-v2 control task from the Box2d simulator within OpenAI Gym. The primary goal of this task is to navigate a lander to its landing pad, with the environment being considered solved at a total reward of 200. For this environment, HDCA and ES trained identical policies represented by a neural network with a hidden layer consisting of 100 neurons, using tanh activation, with 1,000 total parameters. ARS trained a linear policy to stay consistent with the work produced by Mania et al. (2018), requiring 16 parameters to be optimized.

**Effect of sub-problem dimensionality.** This section aims to explore how HDCA's training performance varies with respect to the block size $b$ of perturbed coordinates. Figure 1 shows the total expected reward as a function of the training iteration across five independent trials. In each trial, 5,000 directions were sampled per iteration, corresponding to 5,000 perturbed policies to evaluate. The generated plots demonstrate the maximum, minimum, and average expected reward for all three methods. The subfigures of Figure 1 represent training results for varying coordinate block sizes for HDCA. For example, Figure 1c shows the training performance of HDCA when a block of 50 coordinates is chosen to be perturbed at each iteration. Since ES and ARS perturb all policy parameters when sampling new directions, the number of coordinates perturbed in each subplot of Figure 1 only applies to HDCA.

Table 1: Average number of training iterations before solving LunarLanderContinuous-v2

| Method | Average Training Iterations |
|---|---|
| HDCA-3 | $14.8 \pm 10.5$ |
| HDCA-10 | $4.4 \pm 0.80$ |
| HDCA-50 | $4.6 \pm 2.33$ |
| HDCA-100 | $5.2 \pm 2.99$ |
| HDCA (Average) | $7.25 \pm 4.37$ |

The results from Figure 1 for HDCA are summarized in Table 1 and demonstrate how HDCA can quickly converge to a solution. On average, it took ES 18.4 iterations, ARS 20.8 iterations, and HDCA 7.25 iterations to converge. The number of training iterations was chosen as a primary metric for analysis due to HDCA's scalability with the number of cores. Depending on how HDCA is implemented and the total number of available processors, all function evaluations for a given iteration can be performed in parallel. Thus, looking at how these methods perform relative to the

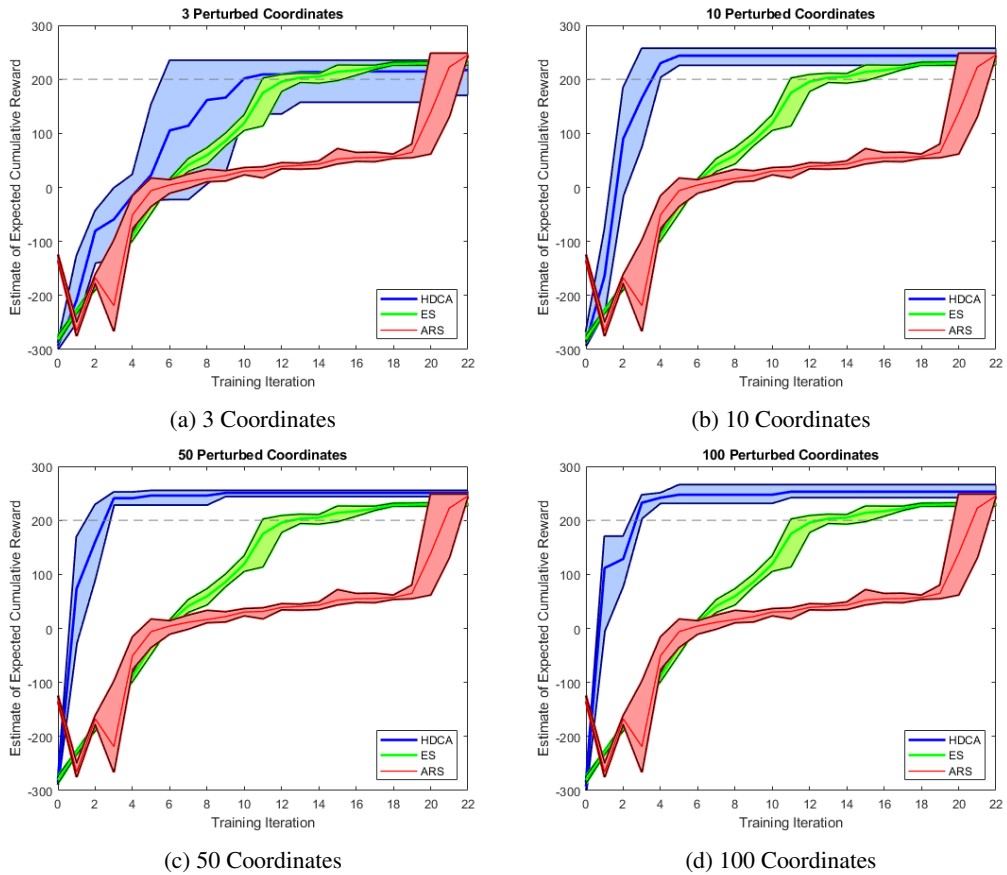

Figure 1: Training performance for HDCA compared to ARS and ES for the LunarLanderContinuous-v2 environment. The estimated expected cumulative reward is plotted against the training iteration, where training continued until the expected reward minus its standard deviation exceeded the threshold of 200. HDCA and ES were trained on a neural network with one hidden layer consisting of 100 neurons, while ARS trained a linear policy. 5,000 directions were evaluated at each training iteration for all three methods, and the number of coordinates perturbed in each subplot refers to the block size $b$ for HDCA.

number of training episodes gives insight into HDCA's viability as an optimization technique as large-scale computing clusters become more accessible.

Following (Salimans et al., 2017), examining the number of training iterations is more meaningful for BBO techniques than sampling efficiency since better parallelization in policy optimizers shall lead to faster training. Furthermore, given that BBO techniques are primarily applied to simulated environments, policy rollouts are much less expensive than in real-world experiments. The primary bottleneck to training is therefore the wall-clock training time. Since every rollout can be executed in parallel for HDCA, this is directly proportional to the number of training iterations assuming adequate resources are available.

All variants of HDCA outperformed ARS and ES in terms of the expected number of iterations until solved besides HDCA-3, which performed consistently with the other methods. The decreased performance of HDCA-3 is likely due to the small overall percentage of network parameters that are perturbed at each iteration. Perturbing just .3% of the total parameters at each iteration, HDCA-3 required more training iterations than the other variants to optimize enough of the model to converge to a solution. Conversely, HDCA-100 required the second most training iterations out of the HDCA variants evaluated. This method perturbed 100 coordinates at each iteration, corresponding to 10% of the total model parameters. However, searching in this high-dimensional space is a much harder task than sub-problems of lower dimensionality, thus performance was observed to be less consis-

tent. To improve training as the sub-problem complexity increases, more samples must be taken at each iteration. Hence, we see a trade-off when increasing block size and find that 'middle-ground' configurations work best. Overall, HDCA proves robust to the choice of block size and demonstrates flexibility and efficient convergence for this task.

**Effect of perturbations per iteration.** Figure 2 demonstrates the relationship between the number of perturbations sampled per iteration and the expected number of iterations until a solution is found. HDCA-10 was chosen for demonstration purposes based on the findings in Table 1. For HDCA, the number of perturbations per iteration represents how extensively the parameter space is explored. In a break from the common paradigm, HDCA does not use its perturbations in a weighted sum but performs a *maximum*-based update from perturbations, which potentially yields much better updates as we increase the number of perturbations. In contrast, ARS and ES base their policy updates on averages across perturbed policy performance, thus the contribution from each perturbation decreases as the sample size increases, and we get diminishing returns due to the concentration of empirical means. This is the primary explanation for why HDCA greatly reduces the number of training iterations with a larger perturbation sample size, while ARS and ES see more similar performance. These results show that HDCA makes better use of increased computational resources.

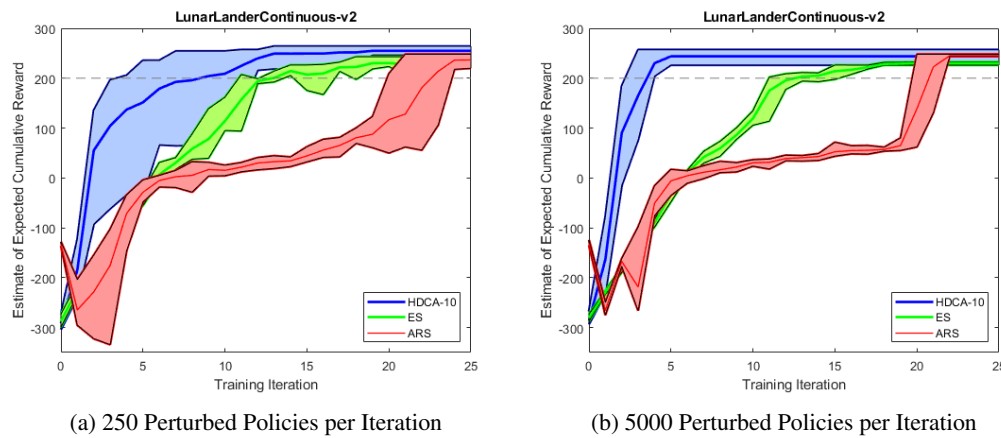

(a) 250 Perturbed Policies per Iteration      (b) 5000 Perturbed Policies per Iteration

Figure 2: Training performance for HDCA-10 compared to ARS and ES for the LunarLanderContinuous-v2 environment. The estimated expected cumulative reward is plotted against the training iteration, where training continued until the expected reward minus its standard deviation exceeded the threshold of 200. The left figure shows the expected number of training iterations when 250 samples are taken at each iteration, and the right shows the same relationship for 5000 samples.

## 4.2 MuJoCo

Next, HDCA was evaluated against a subset of the MuJoCo locomotion tasks, where its training performance was compared to ARS and GLD. All three methods trained identical linear policies. The documented success of linear models (Mania et al., 2018) in the MuJoCo control tasks is a compelling indication that deep neural networks with many parameters may not be necessary for complex RL problems.

The results presented in Table 2 reflect similar trends as those discussed in Sec. 4.1. Swimmer-v2 is regarded as the simplest of the three evaluated MuJoCo environments, and HDCA is rapidly able to converge to an optimal solution. GLD demonstrates similar performance, while ARS is slower to converge. This difference in behavior is likely due to HDCA and GLD's independence of gradient estimates in their policy update steps.

Despite demonstrating similar performance in the Swimmer-v2 environment, HDCA was able to converge to an optimal solution much more quickly than GLD for Hopper-v2. A likely explanation to this divergence is in how the two methods handle stochasticity within the environment. By esti-

mating the numerical mean and variance of policy rewards, HDCA can leverage hypothesis testing to effectively guide its policy updates. Taking a simple maximum of policy rewards ignores how cumulative rewards are a stochastic quantity, which may lead to the slower convergence observed in GLD in more complex domains.

The results for HalfCheetah-v2 highlight the importance of the number of perturbations per iteration. Here we see that HDCA remains competitive with ARS. While HDCA and ARS were able train policies for HalfCheetah-v2 using 500 perturbed policies per iteration, GLD was unable to converge to an optimal solution unless approximately 10x more policy perturbations were generated.

Table 2: Average number of training iterations before solving various MuJoCo locomotion tasks

| Task (Threshold) | Method | Average Training Iterations |
|---|---|---|
| Swimmer-v2 (325) | | |
| | HDCA | $2.2 \pm 0.75$ |
| | ARS | $25.6 \pm 2.65$ |
| | GLD | $2.2 \pm .4$ |
| Hopper-v2 (3120) | | |
| | HDCA | $80.8 \pm 30.9$ |
| | ARS | $121 \pm 34.9$ |
| | GLD | $393 \pm 245$ |
| HalfCheetah-v2 (3430) | | |
| | HDCA | $116 \pm 72.8$ |
| | ARS | $93.6 \pm 11.3$ |
| | GLD | * |

## 5 CONCLUSION

This work develops a gradient-free, black box optimization technique that can match or outperform similar state-of-the-art methods. Its independence from gradients makes it highly scalable and competitive in parallel architectures. Furthermore, its intuitive hyperparameters and simple structure lead to a straightforward implementation that can be easily adapted to other RL environments.

We develop HDCA off of two major insights. Our first insight is to frame policy optimization via coordinate ascent. Coordinate ascent naturally decomposes the problem of optimizing the large policy parameter space into a sequence of smaller, more feasible sub-problems. Although we do not have access to analytical solutions of the sub-problems (as is typical in coordinate ascent), we note that if sub-problems are of a low enough dimension, then any simple numerical optimization strategy shall approximate solutions well. Our second insight is to note that the cumulative rewards, the expectation of which is our optimization criterion, is a stochastic quantity. Thus, the expectation is *estimated* with finite samples of rollouts, which leads to variance and uncertainty about true expectations of cumulative rewards for candidate perturbations. The stochastic nature (and finite sample estimation) of the cumulative reward criterion *has been largely ignored by previous BBO approaches*. Here, we develop a principled approach that accounts for this with one-sided, two-sample hypothesis tests to update policy parameters based on statistical significance. The resulting algorithm is easy to implement, easy to parallelize, and highly effective.

Future work shall explore extensions to HDCA for applications in large network-based policies. Although recent work (Mania et al., 2018) has shown that simple policies are competitive even for complicated tasks, it would be beneficial to leverage larger capacity networks when necessary. (Salimans et al., 2017) showed that using virtual batch normalization (Salimans et al., 2016) allowed for better optimization of larger networks. Thus, we shall explore similar strategies as well as network pre-training schemes for better leveraging of the coordinate ascent block updates when parameters are large.

ETHICS STATEMENT

While all experiments conducted in this work were simulated through software for demonstration purposes, there are many physical problems that can lead to significant consequences if proper safety measures are not considered, such as when developing autonomous vehicles. Therefore, it is imperative to properly define reward functions for environments that could have societal impact. It is equally important to verify that deployed policies accurately reflect this definition. To mitigate the risk of adverse consequences, trained policies should undergo significant testing before deployment.

REPRODUCIBILITY STATEMENT

Our implementation of HDCA is provided in the supplementary material within the "HDCA" directory. Please refer to the README file for execution instructions.

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

## A  APPENDIX

Hypothesis-test Driven Coordinate Ascent (HDCA) combines the insights from sections 2.2 and 2.3, and is presented in Algorithm 3.

**Input:** Initial policy parameters $\theta^0 \in \mathbb{R}^d$;
**Hyperparameters**: Search range $s$, number of noise samples $n$, Number of coordinates per
  block $c$, number of rollouts in mean calculation $M$, hypothesis test parameters $(\alpha, N)$;
Set $i \leftarrow 0$;
**repeat**
  Let $\bar{\mathbf{R}}(\theta^{(i)}), \sigma_{\mathbf{R}(\theta^{(i)})}$ be the empirical average, standard deviation of the current policy
    cumulative reward;
  Let coordinate block $b$ consist of $c$ randomly selected coordinates from the set $\{1, ..., d\}$;
  Sample $\epsilon_{j,k} \sim \mathrm{U}(s)$ **for** $j = 1, ..., n, k \in b$;
  Let $\hat{\theta}^{(j)}$ be the $j^{\text{th}}$ candidate optimizer over block $b$ with values:

$$\forall q \in b^c,\ \hat{\theta}_q^{(j)} \leftarrow \theta_q^i, \quad \forall k \in b,\ \hat{\theta}_k^{(j)} \leftarrow \theta_k^i + \epsilon_{j,k}$$

  Using the policy samples $a_i \sim \pi_{\hat{\theta}^{(j)}}(s_i)$ and environment samples $s_{i+1} \sim p(s_{i+1} \mid s_i, a_i)$,
    sample $M$ rollouts $t_m^{(j)}$, and compute respective cumulative rewards $\mathbf{R}(t_m^{(j)})$,
  $\forall m \in \{1, \ldots, M\}$:

$$t_m^{(j)} \leftarrow \left\{ (s_{m,0}^{(j)}, a_{m,0}^{(j)}), (s_{m,1}^{(j)}, a_{m,1}^{(j)}), \ldots \right\} \quad \mathbf{R}(t_m^{(j)}) = \sum_{(s,a) \in t_m^{(j)}} R(s, a) \qquad (13)$$

  Estimate empirical average cumulative rewards $\bar{\mathbf{R}}(\hat{\theta}^{(j)})$ and corresponding standard
    deviation $\sigma_{\mathbf{R}(\hat{\theta}^{(j)})}$ as:

$$\bar{\mathbf{R}}(\hat{\theta}^{(j)}) \leftarrow \frac{1}{M} \sum_{m=1}^{M} \mathbf{R}(t_m^{(j)}), \quad \sigma_{\mathbf{R}(\hat{\theta}^{(j)})} \leftarrow \frac{1}{M} \sum_{m=1}^{M} (\mathbf{R}(t_m^{(j)}) - \bar{\mathbf{R}}(\hat{\theta}^{(j)}))^2 \qquad (14)$$

  Compute test-statistics $T_j$ for two-sample t-Test to compare each cumulative reward with
    the previous empirical average;

$$T_j = \frac{\bar{\mathbf{R}}(\hat{\theta}^{(j)}) - \bar{\mathbf{R}}(\theta^{(i)})}{\sqrt{\sigma_{\mathbf{R}(\hat{\theta}^{(j)})}^2 / M + \sigma_{\mathbf{R}(\theta^i)}^2 / M}} \qquad (15)$$

  Determine critical value for two-sample t-Test $t_{(j, 1-\alpha, \nu)}$ where $\nu$ is calculated as in Eq. 9 ;
  Determine which policy corresponds to the most statistically significant cumulative reward:

$$\mathbf{j} \leftarrow \arg\max_j T_j - t_{(1-\alpha, \nu_j)}, \quad \textit{diff} \leftarrow T_j - t_{(1-\alpha, \nu_j)} \qquad (16)$$

  **if** *diff ¿ 0* **then**
  | $\theta^{i+1} \leftarrow \hat{\theta}^{(\mathbf{j})}$
  **else**
  | $\theta^{i+1} \leftarrow \theta^i$
  **end**
  $i \leftarrow i + 1$;
**until** *termination test satisfied*;

**Algorithm 3:** Hypothesis-test Driven Coordinate Ascent

