# OpenReview forum: "Hypothesis Driven Coordinate Ascent for Reinforcement Learning"
_ICLR.cc/2022/Conference — ICLR 2022 Submitted_

### Official Review · Reviewer_Je5f · 2021-10-25

**Correctness:** 3
**Technical Novelty And Significance:** 2
**Empirical Novelty And Significance:** 2
**Recommendation:** 5
**Confidence:** 4

**Main Review:**

While the algorithm designed in this paper sounds interesting, it does not appear to be sufficiently novel. In fact, random search methods for black-box optimization have been studied for many years. Especially, in the evolutionary computation community, various approaches to improve the search performance, including block-based search and statistical testing techniques, have been explored extensively in the past to solve various numerical optimization problems. In comparison to existing EC and local search algorithms, it is not clear which part of the new algorithm is truly novel.

The authors claim in the paper that reinforcement learning problems are black-box optimization problems. On such problems, critical gradient information is hard to obtain. However, the prominent success of many state-of-the-art deep reinforcement learning algorithms, such as TD3 and SAC, have already demonstrated the effectiveness of calculating and using gradients to train policies. In comparison to these deep reinforcement learning algorithms, it is not clear what key advantages the new algorithm has. Furthermore, any possible advantages of the new algorithm should be verified both experimentally and theoretically. In particular, the theoretical strength of adopting a gradient-free approach for reinforcement learning is not clearly justified in the paper.

The experimental analysis in the paper has some limitations. First, the number of benchmark problems utilized in the experimental study seems to be quite limited. It is not easy to draw any solid conclusion based on the results obtained from these benchmark problems. Second, the authors argued that the number of training iterations is more meaningful as a performance metric because the algorithms are only applied to simulated environments. Does that mean that the new algorithm cannot be applied to any real-world problems? If so, what is the practical value of the new algorithm? Even when the environment is simulated, it is still important to determine the final learning performance achievable by the new algorithm. For the Mujoco benchmarks, I cannot find any performance results in the main body of the paper. In fact, the authors claimed that the new algorithm can converge to the optimal solution on Swimmer-v2. However, it is not clear what the optimal solution is and how the optimal solution is obtained. Third, the hyper-parameter settings used to solve different problems appear to be different. For example, on the lunar lander problem, 5000 directions were sampled per iteration. Different from that, on half cheetah, the number of sampled policies is reduced to 500 per iteration. It is not clear how the change of hyper-parameters is determined and whether the hyper-parameter settings used present a fair comparison among all the competing algorithms.

Finally, it is claimed in the abstract that this paper has the aim to learn robust policies for stochastic environments. However, to my understanding, most of the benchmark problems utilized in the paper are not highly stochastic in nature. Furthermore, the actual robustness of the learned policies is not evaluated in the paper.

**Summary Of The Paper:**

This paper proposed a new random search algorithm for reinforcement learning. This algorithm features a block-based parameter search technique and the use of statistical testing techniques to determine reliably the next-step search direction.

**Summary Of The Review:**

This paper introduced some interesting ideas for gradient-free policy training for reinforcement learning. However, the novelty of the new algorithm design should be clearly highlighted and strongly justified. The theoretical strength of adopting a gradient-free approach for reinforcement learning also requires better justifications. Meanwhile, the empirical evaluation has some limitations.

---

> ### Author Response · Authors · 2021-11-22
> **Response to Reviewer Je5f**
>
> We appreciate the constructive feedback you left and impressions of the paper. As for the MuJoCo benchmarks, results are presented in Table 2, which details the threshold reward value for the environment to be considered “solved”, as well as the number of training iterations required to reach this threshold. In terms of the comparison to other methods, the number of sampled policies always remained equivalent between methods within a given task, even if this number changes between tasks. Comparing based on the number of training iterations was intended to highlight how black-box optimization techniques tend to perform better in highly parallelized environments. It is much easier to create this parallelization if the environment is simulated rather than in the real world, but this certainly does not preclude HDCA from being run in the real world. Additionally, a policy trained in simulation can be evaluated in a physical domain, thus we believe there is practical value in policies trained by the proposed algorithm.
>
> While methods such as TD3 and SAC have demonstrated effectiveness when calculating and using gradients to train policies, these methods are not BBO techniques, thus we did not feel as though it would be fair to compare HDCA to problems solved by these methods.
>
> The abstract has been updated to remove reference to robust policies, thank you for highlighting this.

---

> > ### Comment · Reviewer_Je5f · 2021-11-28
> > **Thank you for your response**
> >
> > Thank the authors for responding to my comments. The response has addressed some of my concerns. However, I still have concerns regarding the technical novelty, the scale of the experiments, and the effectiveness and sample efficiency of the new algorithm design. I therefore will keep my initial score.

---

### Official Review · Reviewer_ADBA · 2021-11-02

**Correctness:** 3
**Technical Novelty And Significance:** 2
**Empirical Novelty And Significance:** 2
**Recommendation:** 5
**Confidence:** 3

**Main Review:**

As the paper notes, there are several conceptual advantages to an approach like HDCA: it is gradient-free, and well suited to parallelism. I believe, however, that aspects of the paper's exposition could be improved: For instance, in the plots that convey the central results, I do not have clarity on what the y-axis is conveying. The y-axis label differs from the caption, which differs further from what I believe the y-axis is actually stating (the average estimate of the expected return). Similarly, core statements are made that I found a bit hard to follow: "the noise in the reward incurred during an RL policy rollout"; what noise is present in the reward? It seems we have assumed that reward functions are deterministic. Is it the noise in the sequence of random variables associated with future rewards along a random trajectory? If so, it would be helpful to be a bit more precise in some of this language.

Additionally, I have two major questions regarding closely related work.
1. I believe block coordinate ascent has been used for policy optimization previously: See "A block coordinate ascent algorithm for mean-variance optimization" by Xie et al. (NeurIPS 2018). Unless I have missed something critical, it seems important that the authors read this paper and understand what HDCA does beyond their proposed algorithm ("Mean-Variance Policy Gradient" or MVP).
2. The methods of this paper bear a resemblance to _high-confidence policy improvement_ methods, too, in which hypothesis tests are conducted on the value of a proposed new policy to ensure that a better policy is found, based on a set of data. For instance, see "High Confidence Policy Improvement" by Thomas, Theocharous, and Ghavamzadeh (2015), who xamine a variety of methods for attaining probabilistic guarantees about improving policy $\pi_i$ to $\pi_{i+1}$. This work and its successors (Chandak et al 2020, for example) seem conceptually quite similar to the method proposed here. I suspect the present paper would be sharpened with a related work section contrasting the proposed approach with that of policy improvement methods.

For these reasons, I presently lean toward rejection. I am willing to increase my score (pending the perspectives of other reviewers) if the paper's exposition can be sharpened and the relation to the above two bodies of work can be made more clear.

**Minor Questions/Comments**
- "with a potentially non-convex reward function": Are reward functions typically non-convex?
- "... that ABCA is able to take better advantage of the benefits...": Take better advantage than what, gradient estimation? If so, it might be useful to state that more explicitly.
- "Given the noise in the reward incurred during an RL policy rollout": I am not sure I follow. What exactly is the source of the noise? I see that the reward function is defined as a deterministic function, $R : \mathcal{S} \times \mathcal{A} \rightarrow \mathbb{R}$.
- To me, the title does not capture the spirit of the paper. I believe "Hypothesis-Test Driven..." would be more suitable. The phrase "hypothesis driven", to me, evokes the idea that a given learning algorithm is incentivized by falsifying candidate hypotheses in some way.
- I believe I do not understand what is being plotted in Figure 1. The y-axis label says "reward", but the caption says "average expected reward", but I believe these are estimates of the average expected cumulative reward. Can you clarify?

**Writing Suggestions**
-  I believe this sentence could be cleaned up by removing "the expectation of which is our optimization criterion": "That is, the cumulative rewards, the expectation of which is our optimization criterion, is a stochastic quantity whose distribution depends on the stochastic policy and environment."
- "the development of an environment" --> "an environment"
- "its expected discounted reward over time" --> "the expected discounted cumulative reward"
- You can delete the following, as it is contained in the definition of $p$: "A fundamental property of MDPs is that the resulting state is only dependent on the current state and is therefore unaffected by the agent’s prior history"
- "In stochastic environments, a solution to an MDP is represented as a policy $\pi_\theta$, encoded by parameters $\theta$, that maps states to a distribution over actions in a way that maximizes accumulated rewards." This could be true of both deterministic and stochastic environments, but the sentence seems to emphasize stochastic environments. Perhaps just remove the initial phrase?

**Summary Of The Paper:**

This paper introduces a new policy optimization method based on coordinate ascent. As the paper notes early on, the core of the proposed method is based on two central insights. The first is that coordinate ascent is a suitable approach for policy optimization as it decomposes the large search space (the policy space) into "a sequence of sub-problems with smaller search spaces". The second notes that the objective is in fact stochastic: The return can be thought of as a random variable distributed according to the mixture of a possibly stochastic control policy and the possibly stochastic transition dynamics of the environment. In contrast, prior black box approaches to optimization have tended to ignore this stochasticity in favor of optimizing for the expected return (the value). To account for this, the paper proposes a one-sided hypothesis test that can inform whether to update the policy based on the significance of a given rollout. Experiments are conducted on a continuous variant of the LunarLander-v2 domain and MuJoCo, contrasting estimates of the average return achieved by policies vs. iterations for the proposed method (HDCA), an evolutionary approach (ES), and Augmented Random Search (ARS). Results support the improvement of the approach over ES and ARS on this domain.

**Summary Of The Review:**

There are aspects of the proposed approach that seem desirable, but I am reluctant about its novelty due to two closely related pieces of work: (1) A Block Coordinate Ascent Algorithm for Mean-Variance Optimization, and (2) High-Confidence Policy Improvement. Additionally, I believe the paper's overall clarity could be improved.

---

> ### Author Response · Authors · 2021-11-22
> **Response to Reviewer ADBA**
>
> Thank you for your notes, especially the recommendation on adding a discussion on the mean-variance optimization papers you suggested. The paper has been updated to include this review, in addition to many of the other writing suggestions you left. You are also correct in your interpretation of the y-axis representing the estimated expected cumulative reward; the plots and captions have been updated to reflect this meaning.

---

> > ### Comment · Reviewer_ADBA · 2021-11-29
> > **Response**
> >
> > I thank the authors for their comment, and for their detail in addressing the concerns raised by the reviews. The new discussion on mean-variance policy gradient, along with many other writing adjustments (improvements to Figure 1, for instance) certainly strengthens the paper. While I do believe that the improvements do in fact improve the paper overall, I unfortunately do not feel strongly enough about this conviction to champion the paper, in light of the consensus to reject presented by the other reviews.

---

### Official Review · Reviewer_gfVu · 2021-11-07

**Correctness:** 4
**Technical Novelty And Significance:** 2
**Empirical Novelty And Significance:** 2
**Recommendation:** 5
**Confidence:** 4

**Main Review:**

The paper is well-written and clear. I like that the algorithm is relatively simple and benefits from scaling up. The authors have done a good job of justifying the design choices. However, the algorithm is not very novel, essentially consisting of a combination of existing techniques.

My main concern about the algorithm is how well it would scale to more difficult environments. In particular, the paper which made the case for evolution strategies (ES) in RL had experiments on the Humanoid task and Atari games, which raises the question of why HDCA does not compare on these tasks. The experiments in this paper only demonstrate HDCA’s operation in scenarios where a relatively small number of parameters (1000 or less) need to be optimized, and the reward is dense.

There are a few things that could make the paper stronger:
* Experiments on a domain with sparse rewards, or using high-dimensional parameter spaces, as these seem like challenging cases for HDCA. The Atari tasks with CNNs are a common testbed with these characteristics.
* Experiments on non-RL tasks. The HDCA algorithm is in principle applicable to any stochastic black-box optimization problem, but the paper limits itself to RL problems. It would be cool to see that the proposed improvements are effective across domains.
* Plotting a scalar measure of the performance (e.g. how many iterations required to reach a particular performance threshold) as a function of the number of perturbations, so that we can see the diminishing returns more clearly.

I also had a small/unimportant question which could be interesting: why does the block (i.e. indices telling which variables are being optimized) have to be fixed within each iteration? I see no reason why you couldn’t re-sample the block along with the perturbation. This could speed convergence if some parameters are unimportant or harder to optimize.

**Summary Of The Paper:**

The paper proposes a derivative-free optimization algorithm which is applicable to stochastic black-box functions. (The target application area is reinforcement learning, but the algorithm is more broadly applicable.) The core elements of the algorithm are
1. Block coordinate ascent: Only a few parameters are optimized at each step, while the others are held fixed. The maximization is performed by random search over the parameters being optimized, so the function evaluations can be performed in parallel.
2. Hypothesis testing: Because of the high variability of Monte Carlo policy evaluation, a one-sided two-sample hypothesis test is performed to determine whether or not the candidate policy is in fact better than the current policy, and the policy is only updated if this is the case.

The proposed algorithm, hypothesis-driven coordinate ascent (HDCA), is compared to previous derivative-free optimizers proposed in the RL literature, where it is shown to require fewer iterations to converge on most tasks.

**Summary Of The Review:**

The proposed algorithm makes intuitive sense and performs well on the tasks presented by the authors, but it is not particularly novel, and I have doubts about whether it can solve more difficult tasks. Further experiments could change my evaluation.

---

> ### Author Response · Authors · 2021-11-22
> **Response to Reviewer gfVu**
>
> Please see our general comments for a discussion on how our approach separates itself from similarly proposed methods. You are also correct that the block can be re-sampled along with the perturbation at each iteration. However, our incorporation of block-coordinate ascent is to decrease the search-complexity for finding new policies by decomposing the problem into lower dimensional sub-problems. We want to remain faithful to a block-based coordinate ascent approach, and a random search over coordinates and their values is more akin to random search in all parameters. We felt that this conflicted with our inspiration for utilizing coordinate ascent, and thus was omitted from our proposed algorithm.

---

### Official Review · Reviewer_pTGV · 2021-11-08

**Correctness:** 2
**Technical Novelty And Significance:** 2
**Empirical Novelty And Significance:** 2
**Recommendation:** 3
**Confidence:** 3

**Main Review:**

I found the paper to be clearly presented and reasonably well written.

The authors present a simple idea for a new approach for a 'black box optimization' method for learning RL policies. In so doing they contribute to a growing literature noting the extent to which simple optimization methods and linear policies are able to achieve comparable results to current SOTA methods on many RL benchmarks. This work is critical to the RL community.

The new method presented contains a number of magic numbers however (the number of alternate policies-permutations to be considered at each step, the number of steps each policy must be rolled forward, the number of times each policy is to be rolled forward, and the 'significance level' /alpha). To my mind there is simply insufficient experimental evidence to assert the claims made in the paper. Certainly the results look promising on the domains present, but they are few in number. Additionally I do not believe the authors provide for how practitioners might assign these in practice. The authors themselves note, for example, that /alpha is like a "problem specific parameter" but do not make mention of the sensitivity to that parameter in the experiments.

Overall I think the authors have presented an interesting idea - one that finds itself in good company in the literature. But ultimate do not present sufficient evidence to support their more general claims.

nit: if the experimentation presented for a new optimization approach is only to be run against linear policies or single-layer networks that ought to be called out early, perhaps even in the title.

**Summary Of The Paper:**

The authors propose a new 'black box optimization' method for learning RL policies in stochastic environments. The simple new method, called Hypothesis Driven Coordinate Ascent (HDCA), does not explicitly require a gradient estimation, and proceeds instead by rolling forward multiple candidate policies in parallel and taking as best performing policy parameters as the iterate update.

**Summary Of The Review:**

The authors propose a simple new 'black box optimization' method for learning RL policies in stochastic environments. The method is simple and clearly explained. The paper is clearly written and reasonably presented.

This method contributes to a growing literature noting the effectiveness of simple optimization method and linear policies on many current RL benchmarks. However, unlikely notable previous working in this space (in particular, "Horia Mania, et al. Simple random search of static linear policies is competitive for reinforcement learning" who's work this paper references a few times), this paper presents only a limited experimental and sensitivity analysis.

I think the paper would benefit strongly from a more robust experimental treatment of the idea. Specifically, the authors might explicitly restrict themselves to linear policies and do a like-for-like experimental comparison against previous work.

---

> ### Author Response · Authors · 2021-11-22
> **Response to Reviewer pTGV**
>
> Please see our general comments for a discussion on how our approach separates itself from similarly proposed methods. As for your concern on the quantity of magic numbers, we feel as though they have relatively straightforward interpretations that should not require expert fine-tuning.
>
> For example, the number of alternate policies tested at each iteration, as well as the number of steps and how many times a policy should be rolled-forward, are all largely dependent on the available computational resources. The primary purpose for a block-coordinate approach is to decrease the search-complexity for finding new policies, thus increasing the number of perturbed policies essentially increases how densely this search space gets explored. Similarly, increasing policy rollout characteristics improves the approximations used in the hypothesis testing framework.
>
> Additionally, a possible interpretation of the significance level for this approach is the willingness to update the policy to a worse-performing model. This risk-aversion is highly dependent on the task at hand, and we feel as though it is up to the practitioner’s best judgement on what that value should be.

---

### Author Response · Authors · 2021-11-22
**Statement from the Authors**

We would like to thank the reviewers for their time and helpful notes. We believe that Reviewer Je5f captures the essence of our work by stating that our method “features a block-based parameter search technique and the use of statistical testing techniques to determine reliably the next-step search direction.” We feel as though the reliability aspects of our approach add great value to the black-box optimization (BBO) domain, both independently and by permitting a gradient-free policy improvement strategy. Hence, we hope that the reviewers will recommend the article for publication.

Please find general comments and reviewer specific replies below.

We want to emphasize that the novelty in our approach stems from the combination of utilized techniques, which permits a BBO method with a unique set of attractive qualities. As Reviewer Je5f notes, “critical gradient information is hard to obtain” when reinforcement learning problems are black-box optimization problems. Thus, an approach entirely independent of gradient information seems like a valuable tool for solving these problems, especially when used in conjunction with the reliability benefits incorporated in the statistical testing techniques. When compared to similar BBO techniques, we found that HDCA consistently converges in fewer iterations, a significant metric when comparing such methods due to their inherently high degree of parallelism. Additionally, as Reviewer #2 notes, the method is theoretically agnostic to the type of problem being solved, i.e. it can be applied to non-RL domains with minimal changes.

---

### Decision · Program_Chairs · 2022-01-20

**Decision:**

Reject

**Comment:**

The manuscript proposes a black box optimization algorithm based on statistical hypothesis testing and proposes to use it to solve control problems posed in stochastic environments. While "Reinforcement Learning" appears in the title, this appears to have nothing to do with RL other than that it purports to solve problems that are typically used to benchmark RL algorithms, i.e. sequential decision making for reward maximization with stochastic dynamics. Baselines compared against also don't involve the reinforcement learning formalism. A key claim appears to be that the application of other gradient-free methods to such problems incorrectly ignores stochasticity of the objective function owing to the stochastic sequential dynamics.

Reviewers found the paper both clear and well-written, but questioned the novelty of the approach, its situation in the wider world of gradient-free black box optimization methods, and the limited scope of the empirical results. Reviewer pTGV criticized apparent "magic numbers" in the algorithm description, to which the authors responded rather dismissively.

In their general reply statement, the authors seem to misinterpret reviewer Je5f's comment that “critical gradient information is hard to obtain”: my own interpretation is that Je5f is characterizing _black box optimization problems_ in general, and noting that the special structure of MDPs admits gradient estimation via the policy gradient theorem, etc. In response to Je5f directly, the authors state "[TD3 & SAC] are not BBO techniques, thus we did not feel as though it would be fair to compare HDCA to problems solved by these methods". ADBA raises some concerns about links to the existing literature, which appear to have been addressed to their satisfaction, though was unwilling to champion the paper for acceptance in light of the concerns of other reviewers.

The AC concurs with the reviewers' assessments regarding the limited nature of the experiments  The paper makes the claim that the newly proposed method is a compelling choice for the solution of stochastic MDPs, but fails to adequately defend this claim, and fails to benchmark against either policy or value-based RL methods. When challenged on this, they assert that such comparisons would be unfair; this may be true in a limited sense, but if an RL algorithm can outperform the novel BBO method on all problems considered, it's unclear why this would be of significant interest to the ICLR community. A recurring theme in the discussion is that BBO methods are more scalable than RL, though batched A2C methods (e.g. Espeholt et al, 2018) or distributed Q-learning (e.g. Kapturowski et al, 2019) would at least merit mention, and would appear to at least challenge this claim.

Finally, the AC would like to note that work on Evolution Strategies predates the work of Salimans et al, 2017, including for control problems (e.g. Cardamone et al, 2009) and in particular Wierstra et al (2014) propose many sophisticated techniques for more effectively utilizing function evaluations (including methods based on hypothesis testing) which are a natural point of comparison for this work. Similarly, CMA-ES is a widely used and well-regarded technique for gradient-free optimization, and would be a reasonable baseline. In short, the empirical investigation is inadequate even if restricted to gradient-free methods, and the fact that these methods may not explicitly account for stochastic dynamics does not obviate the need for comparison.

The method presented is interesting, and I encourage the authors to continue studying it, with a more diverse set of environments, strong baselines and careful ablations.